# Measurement and Processing of Thermographic Data of Passing Persons for Epidemiological Purposes

**DOI:** 10.3390/s23062945

**Published:** 2023-03-08

**Authors:** Jiří Tesař, Lukáš Muzika, Jiří Skála, Tomáš Kohlschütter, Milan Honner

**Affiliations:** New Technologies-Research Centre, University of West Bohemia, 301 00 Pilsen, Czech Republic

**Keywords:** infrared sensor, temperature measurement of person, IR camera, infrared thermography, artificial intelligence, epidemical screening, public health monitoring

## Abstract

Non-contact temperature measurement of persons during an epidemic is the most preferred measurement option because of the safety of personnel and minimal possibility of spreading infection. The use of infrared (IR) sensors to monitor building entrances for infected persons has seen a major boom between 2020 and 2022 due to the COVID-19 epidemic, but with questionable results. This article does not deal with the precise determination of the temperature of an individual person but focuses on the possibility of using infrared cameras for monitoring the health of the population. The aim is to use large amounts of infrared data from many locations to provide information to epidemiologists so they can have better information about potential outbreaks. This paper focuses on the long-term monitoring of the temperature of passing persons inside public buildings and the search for the most appropriate tools for this purpose and is intended as the first step towards creating a useful tool for epidemiologists. As a classical approach, the identification of persons based on their characteristic temperature values over time throughout the day is used. These results are compared with the results of a method using artificial intelligence (AI) to evaluate temperature from simultaneously acquired infrared images. The advantages and disadvantages of both methods are discussed.

## 1. Introduction

Epidemics can spread rapidly and have significant impacts on public health, thereby it is important to monitor them closely. By monitoring epidemics, public health authorities can identify outbreaks, track the spread of the disease, and implement effective control measures.

Passive surveillance is the most common method for monitoring epidemics, as it relies on healthcare providers to report confirmed cases to public health authorities. This allows public health authorities to track the incidence and distribution of the disease, but it may not detect all cases, as some individuals may not seek medical care or may be misdiagnosed.

Many infectious diseases cause a rise in body temperature. Thereby, temperature monitoring across the population could bring valuable information about the spread of diseases in populations. For such purposes, widespread use of thermographic cameras could be used, as they can measure temperature quickly and easily, without the need for physical contact and thus reducing the risk of cross-infection. This makes them useful in situations where it is necessary to screen a large number of individuals in a short period of time, such as in airports or other public places.

The accuracy of thermographic cameras for the purpose of precise human temperature measurement has been the subject of some debate [1]. Some studies have shown that the cameras can be affected by many factors, e.g., environmental factors, such as ambient temperature and humidity, which can lead to inaccuracies in temperature readings. Additionally, some individuals may have underlying medical conditions that can affect the accuracy of the readings (e.g., inflammation of the eye and cold sores).

Infrared cameras do not allow the measurement of the body’s core temperature, but it is possible to measure the surface temperature. Usually, the area at the corner of the eye (inner canthus) where the temperature is the most stable is measured. It reaches a few degrees lower value than body temperature—somewhere between 33.5 °C and 36.9 °C [2]. The correlation between thermographic measurement of the eye’s maximum temperature and the armpit measurement were presented in [3] (mean value differences up to 0.5 °C, standard deviation of the thermographic measurement of the eyes’ maximum temperature 0.4–0.9 °C). A very strong and positive correlation between the ear temperature (ear contact thermometer) and the forehead temperature measured using high-resolution IR camera was shown in [4]. That suggests forehead temperature could be used for estimation of body’s core temperature.

On the contrary, the review article [5], which is focused on the major issues related to the automatic face recognition and ROI selection for the rapid fever assessment, shows that the maximum temperature from the eye inner canthus seems to be the most reliable method to assess fever [5], more relevant than a forehead temperature examination [3].

The article [6] addresses the choice of ROIs (regions of interest) on the human face for the analysis of the individual’s fever and deals with the temperature thresholds used for this analysis. In [7], it is stated that the traditional thermography method for fever classification (measurements of forehead average temperature and eye maximum temperature greater than 37.5 °C) would not be enough to produce the fever group classification for the proposed CNN algorithm (convolutional neural network). The authors in [8] confirmed that the use of inner canthi temperatures provide clinical approaches for fever screening and in cases of its unavailability the full-face maximum temperatures may provide an effective alternate approach.

The influence of high/low ambient temperature to the temperature of measured people is well known. The experiments in simulated/actual environmental conditions in [9] proved that there must be an acclimatization period ranging from 2 to 9 min depending on the outdoor temperature before temperature screening. The use of relative temperature difference for screening patients with fever is recommended in [10] in the case of ambient temperatures higher than 37.5 °C.

There are many methods of temperature monitoring based on the principle that temperatures higher than the set threshold relate to a possible fever of the investigated person. Such methods and systems were introduced by most IR camera producers in 2020 with the spreading of the COVID-19 disease epidemic [2,11,12,13,14]. In [2] there is the method improved using a moving threshold value affected by environmental influences and previous tested persons. Some systems were also equipped with visible cameras [13]. Many commercial solutions were tested by IPVM (Internet Protocol Video Market) with questionable results [1].

Infrared cameras can be used and are used for enabling/disabling entry into public buildings (hospitals, authorities, factories, etc.) or into a restricted area, e.g., at the airport, etc., as was predicted in 2004 [15]. At all places, the person is measured and declared healthy or sick. His/her data are not evaluated from a statistical point of view or for longer period (weeks, months, etc.).

Thermographic cameras for measuring human temperature have been widely used during pandemics, as it is stated in the overview article [5] and this introduction. The IR camera usage was nonetheless focused mainly on identification of people with higher temperature in certain places. It was not meant as long-term tool for epidemiologists, which could help them predict and control the spread of infectious diseases and protect public health.

This paper focuses on long-term epidemic monitoring using thermographic data and the search for the most appropriate tools for this purpose. The article is intended as the first step towards creating a useful tool for epidemiologists.

## 2. Materials and Methods

This study was focused on the initial verification of thermographic data use for epidemiological purposes. Several places were considered for installation of IR cameras. They had to fulfill following requirements:Passing people should have steady-state temperature.They should have a constant distance from the measuring device.They should not come from the exterior.They should not come from all directions.They should not be bothered by the measurement.The place should be a publicly accessible place.

It should be emphasized that the steady-state temperature in this case is the temperature of the human body under steady-state environmental conditions. Steady-state human body temperature is reached within 30 min according to EN IEC 80601-2-59.

Based on these requirements, several infrared cameras were installed in the pharmacy, the secondary school and the municipal office. This study uses only data from the pharmacy, which is located in the middle of the shopping center. This location meets all the requirements for individual measurements. Since the pharmacy is located in the middle of the mall, persons entering the pharmacy have already been in the mall long enough (~30 min) to reach a steady-state temperature condition. However, in the very limited cases where a person is coming from outside directly into the pharmacy (it takes approximately 5 min to travel from the nearest entrance), the steady-state temperature condition may not be ideal.

This study is intended to serve as an initial insight into what can be expected from the analysis of the temperature of people passing by and what interesting information can be offered for epidemiological purposes.

The results at the end show promising correlations between the measured data and data from general practitioners on people with respiratory diseases.

### 2.1. Epidemiological Data

A number of communicable diseases can pose significant threats at the local, regional or global level that can lead to outbreaks, epidemics or pandemics. Acute respiratory infections (ARIs) are one of the major worldwide health problems associated with high morbidity and mortality [16]. One of the most common pathogens which are associated with ARI are influenza virus (IFV), respiratory syncytial virus (RSV), human rhinovirus (HRV), human parainfluenza virus (HPIV), human adenovirus (HAdV) and human coronavirus (HCoV).

ARIs are a very pressing problem in epidemiology. One of the most common symptoms is a higher body temperature. Therefore, data about ARI were compared with thermographic data. The ARI data were obtained from the local public health office. The health center is informed weekly by general practitioners of the number of patients with ARIs in the prescribed age groups. The health center adds up the figures from all practitioners and recalculates them per 100,000 population. For the purpose of this study the total number of patients in all groups were used for a comparison.

### 2.2. Equipment Description

The Optris Xi 400 infrared camera (Optris GmbH, Berlin, Germany) was used to measure the temperature of people walking by. The IR camera contains an uncooled microbolometric detector with an optical resolution of 382 × 288 pixels. The sensor is sensitive to IR radiation in wavelength range of 7 to 13 μm, which is the most common type of detector used in uncooled IR cameras [17]. The IR camera has an FOV (Field of View) of 18° × 14° of a telescopic lens and temperature range from −20 up to 900 °C divided into three subranges (−20 … 100 °C, 0 … 250 °C and 150 … 900 °C). The range from −20 to 100 °C was used for temperature measurement of persons. The accuracy of the camera is ±2 °C or 2% (greater value), thermal sensitivity (NETD) 80 mK and it can record IR images with a frequency up to 80 Hz/27 Hz.

The selection of this equipment was made due to the very good optical resolution of the detector which, combined with the narrow angle of view, allows measuring the temperature of the human face with high reliability. In addition, if the mass deployment of IR cameras for public health monitoring is required, similar IR cameras are characterized by a very favorable price/performance ratio. These prerequisites make such an IR camera a suitable candidate for mass use, even from an economic point of view.

### 2.3. Place Description

The IR camera was mounted in a pharmacy in the shopping center. It was attached to a rod above a cash register near the ceiling and connected with a communication cable to the control PC in the background of the pharmacy (Figure 1a). The camera’s view was aimed and focused on the automatic entrance door that opens with an incoming person. The measurement location—pharmacy—was selected as a place where people go independently of their health status including normal or higher temperatures (fever). The place inside a shopping center was chosen for the purpose of temperature acclimatization of persons in indoor spaces. In such conditions the temperature measurement of people is much more relevant than measuring a cooled or warmed person, especially in winter or summer months in a freestanding pharmacy with direct access where people come without any acclimatization. In addition, the opening and closing of automatic doors helps to separate people and focus them in one place—the entrance.

### 2.4. Data Acquisition

The thermographic software LabIR developed in our laboratory (NTC UWB) was used for data acquisition. The following parameters of the software were set—emissivity 0.97 [3], reflected ambient temperature 20 °C (there are no objects with high temperature in the pharmacy—no intensive reflections are observed), atmosphere temperature 20 °C and distance 3 m (distance between IR camera and automatic door) and relative humidity 50% for the temperature evaluation. Two rectangular analysis areas are placed in the IR image (thermogram) as shown in Figure 1b. The larger one (R1) covers the area of possible occurrence of human face and the smaller one (R2) the glass door for measurement of surroundings temperature. The IR data are recorded in two ways—the first is in the form of IR images and the second is in the form of continuous recording of temperature values.

The IR images are recorded separately for each person. The software is equipped with an alarm function that enables one to create a condition and act when it is met. Thus, the condition is connected to the maximal temperature of rectangle R1 where the human face can occur, and the threshold temperature is set to 33 °C. The alarm function is only activated when this threshold is exceeded, and the IR image is saved to the file in response to the condition being met. However, the image is not recorded immediately with the alarm function activation, but a time delay of 0.6 s is set to capture the person with the best focus during his/her access to the pharmacy through the automatic door. The new IR image can be taken after alarm function reactivation, so the previous person must be away from region R1. The automatic door helps to capture images in an easier way, as it will obscure the next person entering once it is closed.

Continuous data recording of temperature values does not need any alarm function, it simply saves the time, maximal temperature from region R1 and mean temperature from region R2 into a text file separately for each day. An example of R1 temperature time dependence can be seen in Figure 2. The data sampling is set to “every third frame” of the camera’s natural frequency, which is 27 FPS (frames per second), so 9 temperature values of both regions are saved each second of the entire day. This sampling frequency seems sufficient given the size of the data set and the possibility of identifying a person from the temperature versus time during the evaluation.

### 2.5. Evaluation of IR Images

The evaluation of IR images was performed with the help of AI. Head position (from the front and back) was detected from IR images. Data from the first month were used for training purposes. Altogether, 7648 images were collected. Each head position was then tagged by hand. The individual thermograms were resized to a resolution of 300 × 300 pixels and transformed to greyscale images with fixed temperature ranging from 12 to 42 °C. The obtained images were normalized to take values of 0–1, a standard step that leads to faster solution finding (convergence). To improve model accuracy, additional augmented images were created by shifting, cropping and adding noise, resulting in a total of 23,761 training samples. The network SqueezeNet-SSD [18] was used and trained using Caffe framework [19]. It was trained for 220,000 iterations with a base learning rate of 0.0005. A training loss of 0.61 was achieved. The network provided information about the category—front/back of a head, and further about the size of the head, its location and the calculated confidence of a given category. The result of the proposed procedure is presented in Figure 3 as an example for an (a) incoming and (b) outgoing person.

The maximum temperature from detected regions is then considered to be the temperature of the individual. This procedure can be called face temperature identification (FTI).

### 2.6. Continuous Data Evaluation

A MATLAB script based on temperature identification of persons was developed for separation of each person and determination of their temperatures. The script works in several steps. At the beginning, the input data (text file from continuous data recording) are loaded. The second step is to create a vector of parts of the whole day temperature time dependence where the values are higher than the preset threshold T_lim_. The value of the T_lim_ parameter is chosen with respect to distinguishing between person and no person, so the value T_lim_ = 33 °C was chosen because people with a temperature lower than 33 °C are not expected in a shopping center. These parts must be more than one second away from each other. Otherwise, the part of the vector is not considered as two different people. The third step is to check if the maximal value from the vector part is inside an allowed temperature range that can have a live human being. This range is controlled with limits T_min_limit_ = 33 °C and T_max_limit_ = 40 °C, and these values were chosen for rational reasons. Together with the temperature range check a check of time duration of being in the view of the camera is made that must be longer than 1 s. There is a group of temperatures belonging to separated persons after such procedure that can be called temperature–time identification of persons (TTI). Its functionality is presented in Figure 4, where 14 s of measured temperature–time dependence is shown. The person with a normal temperature is captured in the green oval, a hot object is in the red oval (in this case a cup of coffee) and in the violet oval there is probably a person from behind because of the lower temperature that is above the limit value of 33 °C but its time duration is too short (less than 1 s). The time of capturing of one IR image is highlighted with a red spot (it can be seen that the IR image can be taken after the appearance of the person in the view of the IR camera).

An example of 669 identified persons for one day is shown in Figure 5. The opening hours of the pharmacy from 9 to 21 are clearly seen. The customers’ most exposed times can be recognized around 10 o’clock and between 13 and 18 o’clock when the count of customers exceeds 60 persons per hour. The temperature of 33.7 °C at 7:18 belongs to an employee of the pharmacy.

Evaluation from continuously recorded temperature values has some advantages. First of all, for one person the maximum temperature from his/her time spent in front of the camera can be determined in comparison to the one taken in the IR image. No IR images need to be captured and stored. On the other hand, the main disadvantage is that it cannot be distinguished between temperature of face or neck from behind retroactively.

**Figure 5 sensors-23-02945-f005:**
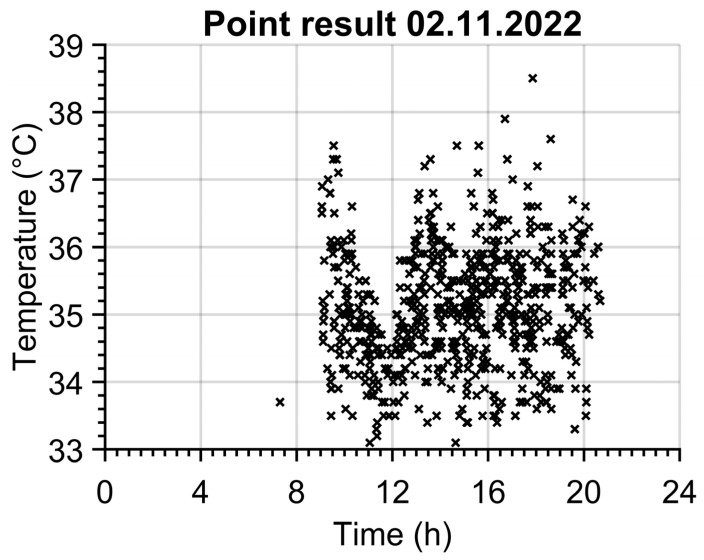
Results of temperature–time identification of persons (TTI) method.

## 3. Results and Discussions

The results show possible ways in which thermographic data for epidemiologic purposes can be used. The different types of evaluation (TTI and FTI) are compared.

The graph in Figure 6 demonstrates differences in measured temperatures by temperature–time identification (TTI) and detection by AI (FTI for face detection and AI-back for back of head detection). In some cases, TTI captured people with their backs to the IR camera. Such data are irrelevant, as they do not contain temperature from faces, which is supposed to correlate the most with the inner temperature of body [8]. As can be seen, the temperature measured from the backside of a person is much lower than the temperature measured from the face area. The temperature obtained from the face is, except for one case, higher than 33 °C. This was a cut off temperature used in TTI to remove non-person data entries, which suggests that the cutoff temperature was selected correctly.

There are also differences between temperatures obtained via TTI and FTI methods. Usually, the difference is tenths of a degree Celsius. The difference is in most cases caused by the non-ideal capturing of a frame for analysis. Basically, the frame is taken after conditions for frame capturing are met. Thereby, a person is not always at an ideal distance from the camera, and thus has little bit lower temperature, since the person is not in total focus of the IR camera. The bigger differences in temperature are caused by the fact that TTI detects a person even though there is no one. This typically happens when there is a hot object which fulfils TTI conditions, e.g., a hot cup of coffee or a hot meal in a box.

In several cases, the hottest place for the person is at the neck. This typically happens in winter season when some people wear scarves outside and they remove them inside. The FTI detects faces without a neck, but the TTI algorithm detects the whole fixed area.

The graph in Figure 7 shows boxplots for TTI and FTI for one month. It is obvious that the FTI average temperature (T_mean_FTI_ = 34.745 °C) is lower than TTI (T_mean_TTI_ = 35.099 °C) and also has a smaller standard deviation (0.851 °C for FTI and 0.908 °C for TTI). Nonetheless, we can notice a correlation in average temperatures which can be better visible in Figure 8. This indicates that when working with average temperatures both algorithms are interchangeable.

**Figure 6 sensors-23-02945-f006:**
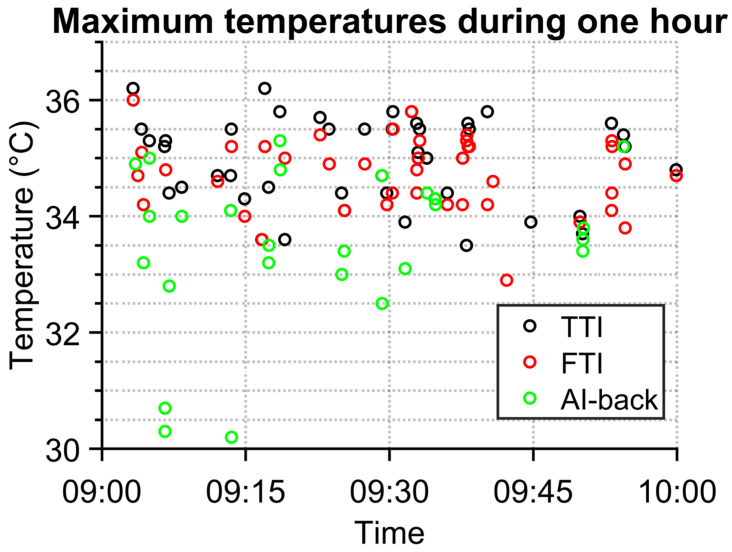
Comparison of one-hour results of TTI and FTI methods.

Significant information about an epidemic can be found in information about how many people visited certain places. Therefore, we compared both FTI and TTI (Figure 9) from the point of view of detected persons per day. TTI detects more people. This is caused by the fact that TTI also detects people from behind and furthermore detects objects which are not people at all.

Typically, the normal human temperature with an armpit thermometer is below 37 °C. Anything above can indicate some kind of illness. Therefore, the epidemic status could be estimated based on how many people had high temperatures. The temperature 37.5 °C was the chosen threshold between a healthy and a sick person. The result in the form of relative number of people with higher temperature is in Figure 10. Neither FTI nor TTI show distinct peaks/trends for the one-month measurement. Nonetheless, the difference between each method is not insignificant. TTI values have a much higher spread and mean value. This may be due to the higher count of identified objects as persons with higher temperatures (as was previously mentioned, this can also be caused by hot objects which are not persons), as shown in Figure 7, where a non-negligible number of high-temperature outliers can be observed.

The results were compared with data from a public health office which collects information from general practitioners. Statistical evaluation in Figure 11 shows a high correlation between data measured with the IR camera (normalized skewness) and normalized data describing acute respiratory infections (ARI) from 22 April 2022 to 2 December 2022. The depicted data were normalized because of the different expressions of data measured by the IR camera and the data from the public health office.

The linear correlation with a Pearson correlation coefficient of 0.78 is depicted in Figure 12. This shows the great potential of using IR screening for epidemiological purposes. It should be pointed out that additional research must be carried out.

The above results are unique because most studies are focused on short time measurements. Currently, temperature screening is the most common usage of IR cameras. They are often used for a one-time temperature measurement of individual persons, e.g., [2] to restrict entry, to select infected persons with elevated temperature for re-measurement by contact method [20] or for the diagnosis of various diseases, such as breast cancer [21].

Long-term monitoring of people’s temperature is not normally carried out, but long-term statistical records of the number of sick people are carried out for epidemic purposes. One of the epidemic threads is an ARI (influenza and others), which is monitored for epidemiologists by general practitioners. Based on epidemic courses and trends, measures are announced and carried out (e.g., ban on visits to hospitals and social facilities and flu holidays) [16].

**Figure 11 sensors-23-02945-f011:**
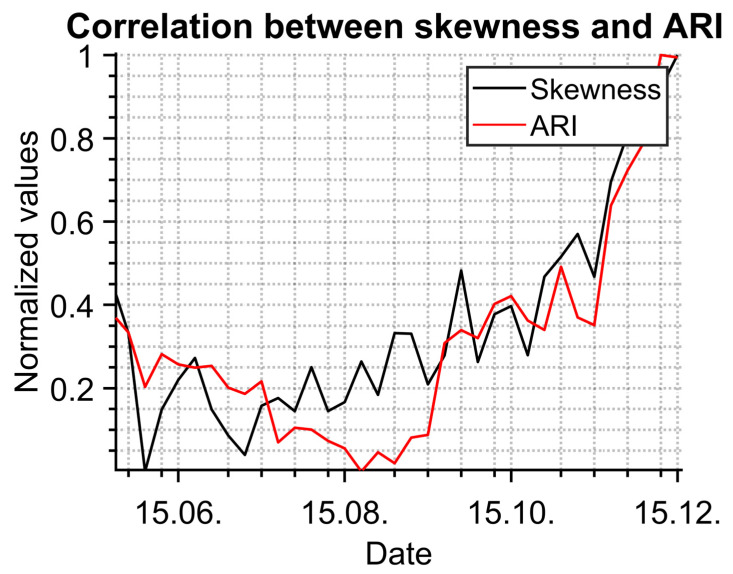
Statistic evaluation of measured people in comparison with data from public health office.

**Figure 12 sensors-23-02945-f012:**
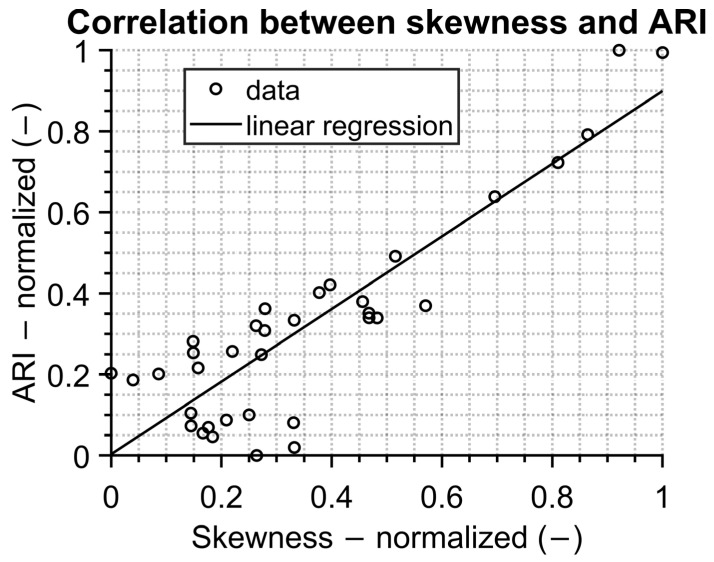
Correlation between statistic evaluation of measured data results and data from public health office.

Many studies [9,22] suggest that the influence of the outdoor environment on the temperature measured by the IR camera has a significant effect. We agree with this, which is why we placed the IR camera in the center of the shopping center to eliminate this influence as much as possible.

## 4. Conclusions

The article deals with the noncontact temperature measurement of passing people using an IR camera. It is focused on the methods of data evaluation and the comparison of the classic approach—temperature–time identification of person (TTI)—with face temperature identification by AI (FTI). The correct use of the thermal sensor (IR camera) is discussed. The selection of appropriate places inside a building enables avoiding heated/cooled people in summer/winter. In addition, the automatic opening door to the pharmacy, where the camera was placed, helps distinguish individual entering people.

The number of identified people and their maximum temperatures obtained by both methods are in good correlation. The mean value of temperature for the TTI method is 0.35 °C higher than the mean temperature resulting from the FTI method, and the values of standard deviations are almost the same for both methods, approximately ±0.9 °C—see Figure 7. The number of people measured with the IR camera identified by the FTI and TTI methods differs on average by 150 persons per day because of the method of evaluation of the single IR images (only a person facing the IR camera works for the FTI method).

The comparison between temperature evaluation and time trend data from the public health office shows very good correlation. The results seem to be interesting for public health purposes to monitor or predict epidemic trends.

Preparations are being made to build the FTI method in a simplified form into the recording software for future work. Our idea is to identify the presence of a person in the view of the IR camera using AI and record only correct single IR images and save only temperature values belonging to that person instead of continuous saving of values throughout the whole day.

Future work in the field will be focused on the correlation between measured temperatures and data obtained from the local public health office. Promising results are presented in this study, but a longer period of measurements is needed to fully validate this approach.

## Figures and Tables

**Figure 1 sensors-23-02945-f001:**
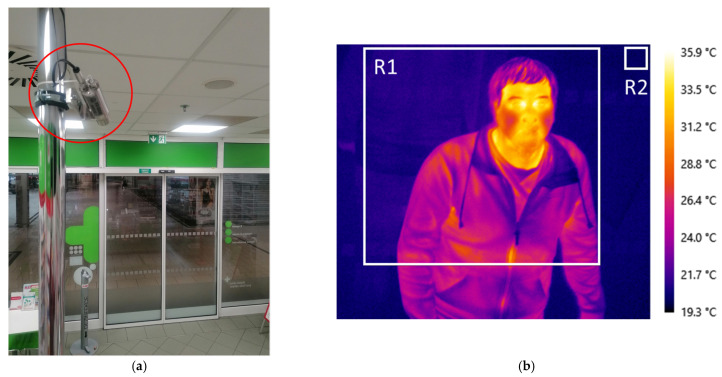
The place of measurement system installation—pharmacy in a shopping center: (**a**) placement of the IR camera (highlighted in red oval) and its view to the automatic entrance doors; (**b**) IR image (thermogram) of passing person (author 1).

**Figure 2 sensors-23-02945-f002:**
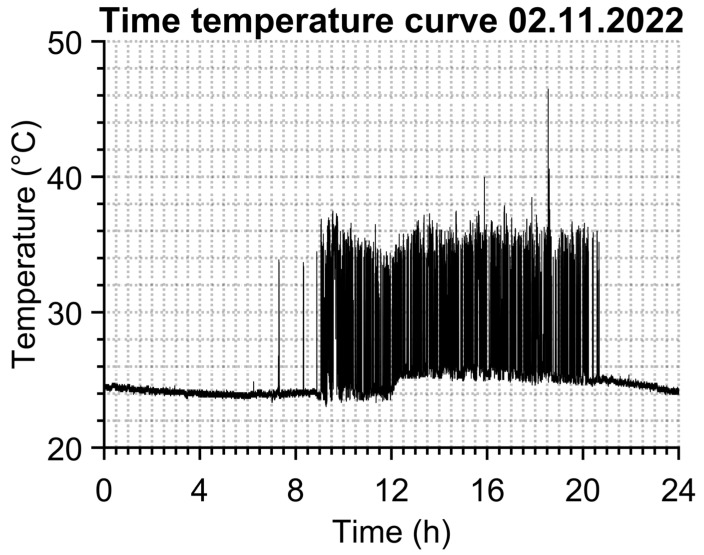
Temperature time dependence for one day.

**Figure 3 sensors-23-02945-f003:**
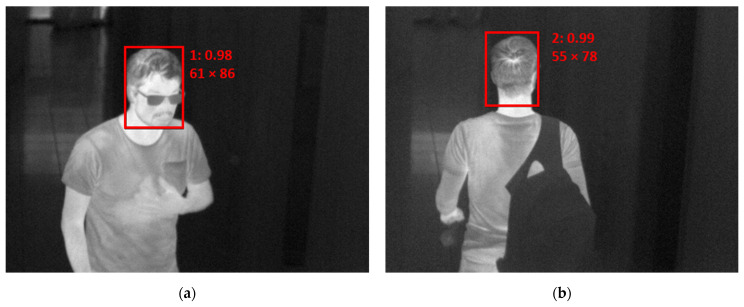
Example of detection by AI: (**a**) head front; (**b**) head back (both pictures is author 2).

**Figure 4 sensors-23-02945-f004:**
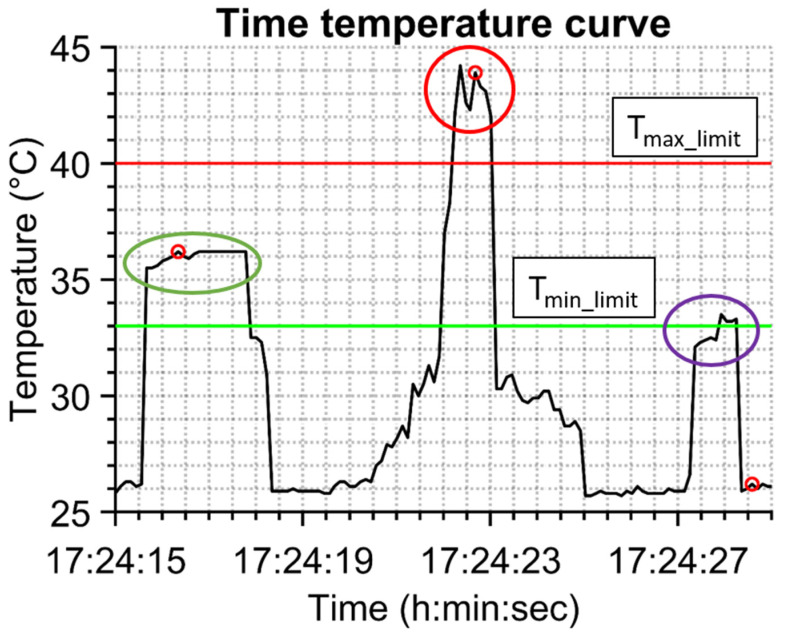
Procedure of temperature–time identification of persons (TTI) method.

**Figure 7 sensors-23-02945-f007:**
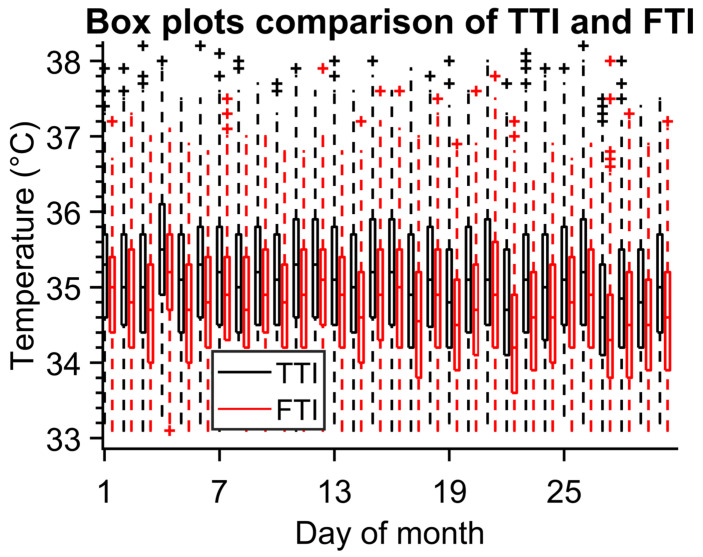
Month values of person temperatures obtained by both methods (TTI and FTI).

**Figure 8 sensors-23-02945-f008:**
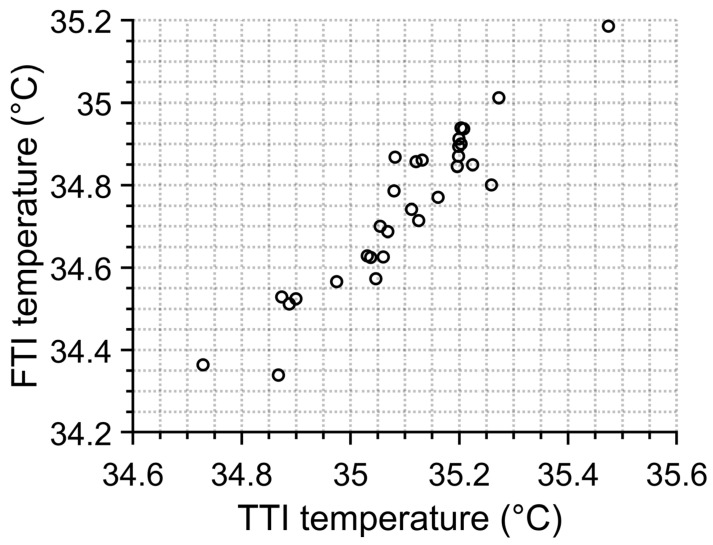
Correlation between average temperatures resulting from FTI and TTI methods.

**Figure 9 sensors-23-02945-f009:**
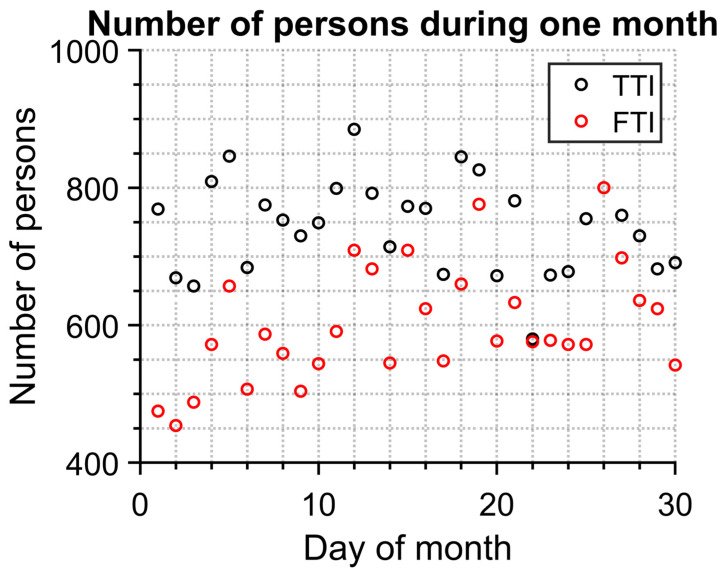
Correlation between methods FTI and TTI—number of persons.

**Figure 10 sensors-23-02945-f010:**
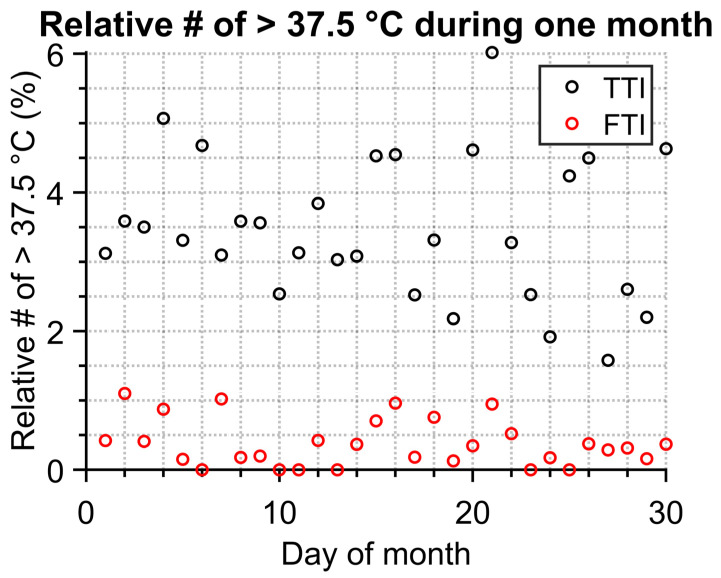
Correlation between FTI and TTI methods—relative count of person with temperature higher than 37.5 °C.

## Data Availability

The data are available upon request to authors.

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
