# Peer review of "Measurement and Processing of Thermographic Data of Passing Persons for Epidemiological Purposes"

_sensors, 2023, doi:10.3390/s23062945_

Round 1

Reviewer 1 Report

This paper presents a study on the measurement and processing of thermographic data of passersby for epidemiological purposes. The study aims to provide insight into predicting epidemic trends using facial skin temperatures of passersby. However, the paper appears to have some flaws as a scientific paper. Please consider the following comments for improvement:

1. The Introduction section needs clarification on the novelty of this study. The authors should provide a clear description of the issues of previous studies, the position of this study, and the purpose of this study. As there are many studies on the estimation of human body temperature using thermography, it is crucial to differentiate this study from previous studies.

2. In line 98, the authors state that passing people should have a steady temperature, but the meaning of "steady" is unclear and abstract. Please provide a concrete explanation to clarify this point.

3. The paper would benefit from a more detailed explanation of how the health center measures the data and how the data is processed. Please include this information in the Materials and Methods section.

4. There is a contradiction between lines 248-249, where TTI is reported to detect a hot object as a person even though there is no person, and lines 277-279, where TTI is said to identify a large number of people at high temperatures. Please address this contradiction.

5. The paper would benefit from a discussion of the results in the context of previous studies. The current discussion does not show how the results of the study relate to and contribute to previous research.

Overall, the study's topic is important and timely, but addressing the above comments will improve the paper's scientific rigor and clarity.

Reviewer 2 Report

The paper "Measurement and Processing of Thermographic Data of Passing Persons for Epidemiological Purposes" reports on statistical data analysis of thermal-IR imaging data acquired on a random sample of population entering a pharmacy shop. IR data have been acquired in two modes: face temperature identification (FTI) and time-temperature dependence. In the first mode, an AI algorithm selects the face of any individual and calculates the average temperature. In the second mode, a time trace of maximum temperature in the image vs. time is acquired per each day, providing a "general trend" of temperature vs. time in the day. In the last part of the paper, both types of data are globally analyzed over more than 6 months so as tohighlight teh correlation of "number of individuals with anomalously high temperature" (measured as skewness of the distribution) vs. some epidemiologic control parameter, and a very good correlation is found.

Alltogether, the research design is very solid, the mathematical and statistical tools seems to be very appropriate, the text is clear and detailed, and figures with images and data are failry well prepared. The conclusions are very honestly presented: measurement of temperature of a single individual is just NOT possible; Statistical trends over one day are not significant: statistical trends over months may be significant and, in the future, they could provide a new epidemic-alert tool. The paper is therefore publishable as is.

I have one single technical comment that may be taken into account for improving the presentation quality, related to the physics of thermal IR emission by solid bodies. The blackbody emission law, or Stefan-Boltzmann law, directly relates the emitted IR radiation power to the temperature of a body. All thermal IR cameras implement this law into their data output, with or without more or less correct calibration procedures. However, the real thing is that the camera not only detects the emissivity, but also the reflected IR power emitted by other bodies in the environment, including walls, doors, ceiling etc. This is why some surfaces of the skin somethimes look "shiny" in a thermal IR image. And this reflection contribution also explains the strong correlation between the environmental temperature at some time of the day observed by the authors. At least, the authors should mention that the effect of reflected IR power adding to emitted IR power has been totally negleted in the  temperature evaluation, as often done in thermal IR imaging, for example when they write:

"The following parameters are set to the software: emissivity 0.97 [P],
reflected ambient temperature 20 °C, atmosphere temperature 20 °C and distance 3 m (distance between IR camera and automatic door), relative humidity 50% for the temperature evaluation."

This sentence clarifies that no reflected IR power was considered, otherwise the environmental temprature should enter the set of parameters (see for example  Sensors 2023, 15(4), 907, where the opposite extreme case is considered: reflection dominating over emission).

Round 2

Reviewer 1 Report

All concerns have been satisfactorily addressed.